# Introducing a Linear Empirical Correlation for Predicting the Mass Heat Capacity of Biomaterials

**DOI:** 10.3390/molecules27196540

**Published:** 2022-10-03

**Authors:** Reza Iranmanesh, Afham Pourahmad, Fardad Faress, Sevil Tutunchian, Mohammad Amin Ariana, Hamed Sadeqi, Saleh Hosseini, Falah Alobaid, Babak Aghel

**Affiliations:** 1Faculty of Civil Engineering, K.N. Toosi University of Technology, Tehran 158754416, Iran; 2Department of Polymer Engineering, Amirkabir University of Technology, Tehran 1591634311, Iran; 3Department of Business, Data Analysis, The University of Texas Rio Grande Valley (UTRGV), Edinburg, TX 78539, USA; 4Energy Institute, Energy Science and Technology Department, Istanbul Technical University, Istanbul 34469, Turkey; 5Department of Petroleum Engineering, Gachsaran Branch, Islamic Azad University, Gachsaran 6387675818, Iran; 6Department of Internet and Wide Network, Iran Industrial Training Center Branch, University of Applied Science and Technology, Tehran 1599665111, Iran; 7Department of Chemical Engineering, University of Larestan, Larestan 7431813115, Iran; 8Institut Energiesysteme und Energietechnik, Technische Universität Darmstadt, Otto-Berndt-Straße 2, 64287 Darmstadt, Germany; 9Department of Chemical Engineering, Faculty of Energy, Kermanshah University of Technology, Kermanshah 6715685420, Iran

**Keywords:** biomass sample, heat capacity, empirical correlation, biomass crystallinity, feature reduction

## Abstract

This study correlated biomass heat capacity (Cp) with the chemistry (sulfur and ash content), crystallinity index, and temperature of various samples. A five-parameter linear correlation predicted 576 biomass Cp samples from four different origins with the absolute average relative deviation (AARD%) of ~1.1%. The proportional reduction in error (REE) approved that ash and sulfur contents only enlarge the correlation and have little effect on the accuracy. Furthermore, the REE showed that the temperature effect on biomass heat capacity was stronger than on the crystallinity index. Consequently, a new three-parameter correlation utilizing crystallinity index and temperature was developed. This model was more straightforward than the five-parameter correlation and provided better predictions (AARD = 0.98%). The proposed three-parameter correlation predicted the heat capacity of four different biomass classes with residual errors between −0.02 to 0.02 J/g∙K. The literature related biomass Cp to temperature using quadratic and linear correlations, and ignored the effect of the chemistry of the samples. These quadratic and linear correlations predicted the biomass Cp of the available database with an AARD of 39.19% and 1.29%, respectively. Our proposed model was the first work incorporating sample chemistry in biomass Cp estimation.

## 1. Introduction

Global warming [1,2] and limitations of fossil fuel sources [3] have been two main problematic issues in recent decades. According to reports, the maximum allowable carbon dioxide (CO_2_) concentration has exceeded 70 ppm in the atmosphere from the preindustrial period [4]. The combustion of coal and petroleum [5], natural gas industries [1], and petrochemical complexes are responsible for 80% of CO_2_ emissions to the atmosphere [6,7]. Furthermore, cement, steel, and iron manufacturers are the subsequent sources of CO_2_ emissions [4]. In this way, significant attention has been paid to carbon capture [8] and sequestration strategies [9] to reduce, control, and utilize greenhouse gases, including CO_2_, methane, nitrogen, sulfur, chlorofluorocarbons, and so on [10,11]. To this end, according to the BLUE map scenario of the international energy agency [12], sustainable energy sources, including biomass [13], biogas [14], and solar energy [15], have been introduced as promising candidates to replace traditional fossil fuels.

Recently, biomass-to-energy processes have received growing interest because of the energy and global warming crises [16]. According to the United Nations Environment Program (UNEP) [17], 140 billion tons of biomass (mainly agricultural and wooden wastes) are produced throughout the world annually [18]. Wide ranges of added-value chemicals and biofuels may be synthesized from this low-cost, sustainable, and plentiful renewable feedstock [19]. A schematic illustration of synthesizing various products from lignocellulosic biomass is presented in Figure 1. Based on UNEP [12], around 20 times the available environmental yield of agricultural production technologies is required to achieve sustainable development in 2040 [20,21]. Accordingly, thermal processes of biomass, such as gasification and pyrolysis, have emerged as practical technologies to convert these types of biomass samples into valuable products [22,23]. It is worth noting that the mentioned processes include during the first pyrolysis step, in which biomass is converted to gas and a solid carbon in the presence of heat [24].

Reliable knowledge of the thermal characteristics of biomass is a crucial issue for biomass-to-energy process design [22]. Indeed, molecular kinetics govern the thermal behavior of biomass valorization processes [22,25]. Numerous experimental/modeling studies have been devoted to the determination of thermal properties of biomass, including elemental composition [26], higher heating value [27], thermal conductivity [28] and specific heat capacity [13].

Biomass heat capacity is often experimentally measured by differential scanning calorimetry (DSC) [29]. Although the DSC is an accurate method, utilizing a few milligrams of sample results in a shallow heat throws doubt on the measurement accuracy [22,24,25]. It is worth noting that measuring biomass heat capacity at a higher temperature than 423 K has some limitations due to sample decomposition [24]. Bitra et al. determined the heat capacity and thermal conductivity of kernels, peanut pods, and shells using a purpose-built vacuum flask calorimeter [30]. Furthermore, Mothée and De Miranda focused on the thermal analysis of sugarcane bagasse and coconut fiber as typical agricultural byproducts [31]. In addition, the heat capacity of cellulose regarding its applications in the pulp industry and tissue engineering has been investigated in many studies [32,33,34]. Ur’yash et al. employed the adiabatic calorimeter method to explain the water effect on biomass heat capacity in temperatures ranging from 80 K to 330 K [34]. Blokhin et al. experimentally measured the heat capacity of biomass samples obtained from four different sources [32].

Generally, laboratory-scale measurements are complicated and time-consuming, require economic expense, and often contain different uncertainty levels associated with human error and the wrong calibrations of apparatus. Furthermore, it is hard to directly incorporate experimentally measured data for computer-aided simulation purposes. Therefore, it is necessary to develop a correlation to estimate biomass heat capacity from some available features. This type of correlation reduces experimental cost, saves time, and can be easily coupled with computer-aided simulators. The heat capacity (Cp) of a wide range of pyrolysis chars obtained under conditions representative of industrial reactors was correlated in a temperature range of 40–80 °C [22]. Kollman and Cote suggested an empirical correlation for estimating the specific heat capacity of solid wood as a function of the temperature valid for 0–100 °C [35]. In another attempt, Gupta et al. proposed a model for softwood barks and their derived softwood chars using differential scanning calorimetry, showing acceptable Cp predictions in the temperature range of 40–140 °C [36]. The mathematical formulas of the literature suggesting correlations for estimating biomass heat capacity have been widely investigated in Section 2.2.

All these correlations only estimate the biomass heat capacity as a function of temperature and ignore the effect of biomass chemistry [22,35,36]. Since biomass chemistry influences heat capacity, it is necessary to include such information in the model’s entry. Accordingly, this study developed a simple correlation to estimate biomass heat capacity by considering the sample’s chemistry and operating conditions. To our knowledge, it is the first usage of the bio-sample composition (sulfur and ash contents), crystallinity index, and temperature to estimate the biomass heat capacity. This study also compared the accuracy of the developed correlation with those suggested in the literature. This type of correlation helped to enhance the understanding of biomass composition on heat capacity. 

## 2. Results and Discussion

Multiple linear regression was employed to estimate the biomass heat capacity from the sample chemistry, crystallinity index, and temperature. Then, the proportional reduction in error was applied to simultaneously decrease the model size and increase its accuracy. Then, the accuracy of the constructed correlation was compared with those suggested in the literature. Finally, several graphical and numerical investigations were performed to monitor the prediction accuracy of the proposed model in real-field situations.

### 2.1. Developed Correlations in This Study

Equation (1) shows a simple linear correlation developed to estimate the heat capacity of biomass from four different origins in a temperature range of 81 to 368 K. The differential evolution algorithm [37] was used to adjust the coefficients of this correlation using 576 experimental datasets.
(1)$$Cp^{cal}\;=\;-\;0.0158\times CI\;-\;0.0011\;\times \;Ash\;\;+\;0.022\;\times \;S\;+\;0.00407\;\times \;T\;+\;0.0165$$

This correlation estimated the biomass heat capacity with the absolute average relative deviation (AARD%) of 1.1%. Equation (2) was used to calculate the AARD% from the experimental ($Cp^{\exp}$) and calculated ($Cp^{cal}$) biomass heat capacities [38,39]. Here, *N* shows the number of available datasets, i.e., 576.
(2)$$AARD\%\;=\;\sum_{i\;=\;1}^{N}100\;\times \;\left(\left|Cp_{i}^{\exp}\;-\;Cp_{i}^{cal}\right|/Cp_{i}^{\exp}\right)/N$$

The proportional reduction in error (PRE) is a practical method for conducting the sensitivity analysis on independent variables and ranking them based on their contribution to the model’s accuracy [40]. The PRE uses the sum of squared errors (SSE) or mean squared errors (MSE) to conduct its duty. Equations (3) and (4) express the mathematical formulation of the SSE and MSE, respectively [41].
(3)$$SSE\;=\;\sum_{i\;=\;1}^{N}{\left(Cp_{i}^{\exp}\;-\;Cp_{i}^{cal}\right)}^{2}$$
(4)$$MSE\;=\;\sqrt{\sum_{i\;=\;1}^{N}{\left(Cp_{i}^{\exp}\;-\;Cp_{i}^{cal}\right)}^{2}/N}$$

The developed correlation (i.e., Equation (1)) has four parts, with each part including one independent variable. The PRE associated with each independent variable was obtained using Equation (4).
(5)$$PRE\;=\;(SSE_{part}\;-\;SSE)/SSE_{part}\;\;\;\;\;\;\;part\;=\;1,\;2,\;3,\;4$$

Here, SSE and $SSE_{part}$ indicate the sum of squared errors obtained by considering all terms (i.e., Equation (1)) and excluding the i^th^ part of the original correlation, respectively. A high PRE shows that a considered independent variable has a more substantial role in model accuracy and vice versa.

Figure 2 represents the PRE related to the crystallinity index, ash and sulfur contents of the bio-samples, and temperature.

This figure justifies that the temperature and crystallinity index are the main parameters that improve the model’s accuracy. On the other hand, the ash and sulfur contents only enlarged the model size and had a low contribution to its accuracy.

The previous analysis approved that it was better not to consider the ash and sulfur contents of the bio-samples in the model development, and re-design the correlation solely based on the temperature and crystallinity index. Equation (5) presents the mathematical shape of the developed correlation utilizing the most important independent variables.
(6)$$Cp^{cal}\;=\;-\;0.0156\times CI\;+\;0.00407\;\times \;T\;+\;0.0162$$

This three-parameter correlation was more straightforward than the previous five-parameter model, needing low entry information, and providing higher accuracy, i.e., AARD = 0.98%.

### 2.2. Comparison with the Literature Suggested Models

It was previously explained that the available correlations in the literature estimated the biomass heat capacity by either quadratic or linear relationships with the temperature. Therefore, it was necessary to adjust the coefficients of Equation (6) utilizing the available experimental database. It is worth noting that *A* equaled zero for the linear correlation.
(7)$$Cp^{cal}\;=\;A\times T^{2}\;+\;B\;\times \;T\;+\;C$$

Table 1 presents the numerical values of the adjustable coefficients of the linear and quadratic correlations based on the temperature. The last column of Table 1 indicates that the linear and quadratic correlations predict the heat capacity of the 576 bio-samples with the AARD of 1.29% and 39.19%, respectively. Not considering the effect of bio-sample chemistry on the heat capacity may have been responsible for this non-logical uncertainty [22,35,36].

### 2.3. Visually Inspecting the Performance of the Three-Parameter Correlation

Up to now, it has been approved that the three-parameter linear correlation based on the crystallinity index and temperature has the highest accurate prediction for biomass heat capacity. This section employs several graphical analyses to inspect the model performance further. The compatibility between experimental and calculated heat capacities for different biomass samples is depicted in Figure 3. It can be concluded that an outstanding level of agreement existed between the laboratory-measured and calculated heat capacities. The developed correlation encountered problems in predicting the fourth class’s biomass heat capacity with excellent accuracy (wood amorphous cellulose).

Since the crystallinity index of this biomass type was zero, the first part of the developed correlation diminished (i.e., Equation (5)). Indeed, the heat capacity of the biomass obtained from wood amorphous cellulose was estimated using a linear correlation solely based on the temperature. This explanation may justify the uncertainty observed in predicting the biomass heat capacity of the fourth class, especially at higher values.

The histogram of the arithmetic deviation between experimental and calculated biomass heat capacities (i.e., residual error) is illustrated in Figure 4. This figure states that the heat capacity of four different biomass classes was estimated with infinitesimal residual errors, mainly between −0.02 to 0.02 J/g∙K. It was interesting to see that 200 biomass heat capacities were calculated with the residual error equaling zero.

Figure 5 shows the AARD observed for predicting the heat capacity of each biomass class. The heat capacity of the first to the fourth biomass classes was estimated with the AARD of 0.86%, 0.53%, 0.83% and 1.71%, respectively. The developed correlation presented the overall AARD = 0.98% for estimating the heat capacity of all 576 bio-samples.

### 2.4. Analyzing the Cp of Bio-Samples with Different Origins

The experimental and modeling profiles of heat capacity versus temperature for the considered biomass classes are plotted in Figure 6a,d. Excellent compatibility between the laboratory measurements and the correlation predictions can be concluded from these figures. Furthermore, it can be seen that biomass heat capacities have a substantial direct relationship with temperature.

It should be highlighted that by reducing the crystallinity index from the first to the last biomass classes, the first term of the developed correlation (i.e., Equation (5)) gradually became weaker. As mentioned previously, the developed correlation finally appeared as a linear model based on the temperature only for the fourth biomass class (i.e., wood amorphous cellulose). The deviation between experimental and calculated heat capacities increased by decreasing the crystallinity index (compare Figure 6a–d). Although the observed deviation level was too small to distort the generalization ability of the developed correlation, it approved the importance of incorporating the sample’s chemistry in the heat capacity estimation phase.

### 2.5. Pair Effect of Temperature and Crystallinity Index on the Biomass Cp

Furthermore, the variation in heat capacity of biomass based on temperature and crystallinity index is plotted in Figure 7. As can be observed, there is a linear relation between employed variables and related heat capacity. Additionally, temperature enhancement has a significant effect on the biomass heat capacity than crystallinity index, which was already proved by PRE analysis (Figure 2).

### 2.6. Summary of the Study in the Flowchart Form

A summary of the steps followed in this study to develop and validate a linear correlation to estimate the biomass heat capacity is graphically introduced in Figure 8.

## 3. Materials and Methods

This section presents experimental biomass heat capacities collected from the literature. Furthermore, the literature suggesting correlations for estimating biomass heat capacity has been reviewed.

### 3.1. Collected Database

This study focused on constructing a correlation to estimate the heat capacity of biomass from four origins, including cotton microcrystalline cellulose (sample 1), wood sulfite cellulose (sample 2), straw cellulose (sample 3), and wood amorphous cellulose (sample 4), were considered [32]. The chemical composition of bio-samples, crystallinity index, and temperature were considered in this estimation. Table 2 introduces numerical ranges of sulfur and ash mass fractions, crystallinity index, temperature, and the associated heat capacity of the considered bio-samples [32]. A multiple regression method was applied to linearly relate bio-sample heat capacity ($Cp^{cal}$) to the ash (*Ash*) and sulfur (*S*) contents, crystallinity index (*CI*), and temperature (*T*) based on Equation (8).
(8)$$Cp^{cal}=\;f\;\left(CI,\;Ash,\;S,\;T\right)$$

It should be mentioned that all constructed correlations in this study were only valid for estimating the heat capacity of those bio-samples with the composition listed in Table 2, in the temperature range of 80.53 to 368.25 K.

### 3.2. Literature Suggested Correlations

As previously mentioned, the literature suggested several correlations for estimating the bio-sample heat capacity. Table 3 summarizes the mathematical formulations of these correlations and the range of applications. This table shows that all the developed correlations only use temperature to estimate heat capacity. Generally, these correlations use quadratic or linear forms to correlate the bio-sample heat capacity to the temperature.

Since these correlations were only valid to estimate the heat capacity of the considered biomass in the specific operating ranges, their coefficients were needed to readjust to cover the utilized database in the current study (see Section 3.2).

## 4. Conclusions

This study aimed to develop an accurate and straightforward correlation for estimating biomass heat capacity, considering bio-sample chemistry and operating condition. The proportional reduction in error justified that temperature and crystallinity index of bio-samples were the most critical factors affecting biomass heat capacity. Multiple linear regression was applied to develop a three-parameter correlation based on the selected features. The proposed model was more accurate than the quadratic and linear models available in the literature. This model predicted 576 biomass heat capacities with the AARD = 0.98%, while the previously suggested models presented higher uncertainty for the same database. The constructed correlation in this study predicted the heat capacity of biomass samples from four different origins with an excellent AARD of lower than 1.71%. The sensitivity analysis approved that the deviation between experimental and calculated heat capacities increased by reducing the crystallinity index of biomass samples. This observation indicated the importance of incorporating biomass chemistry in the estimating phase of the heat capacity. Since this study estimated the heat capacity of biomass samples with low ash content, it is a good idea to consider straw/digestate as the independent variable in future research, where the influence of ash on the specific heat capacity would not be neglectable.

## Figures and Tables

**Figure 1 molecules-27-06540-f001:**
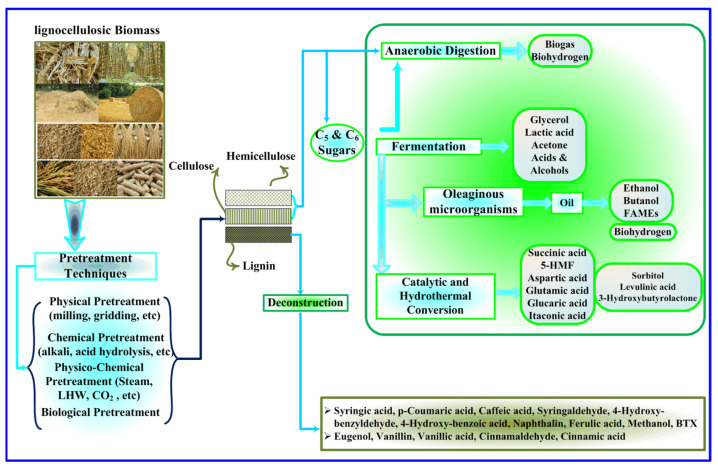
Schematic illustration of various valuable products from lignocellulosic biomass.

**Figure 2 molecules-27-06540-f002:**
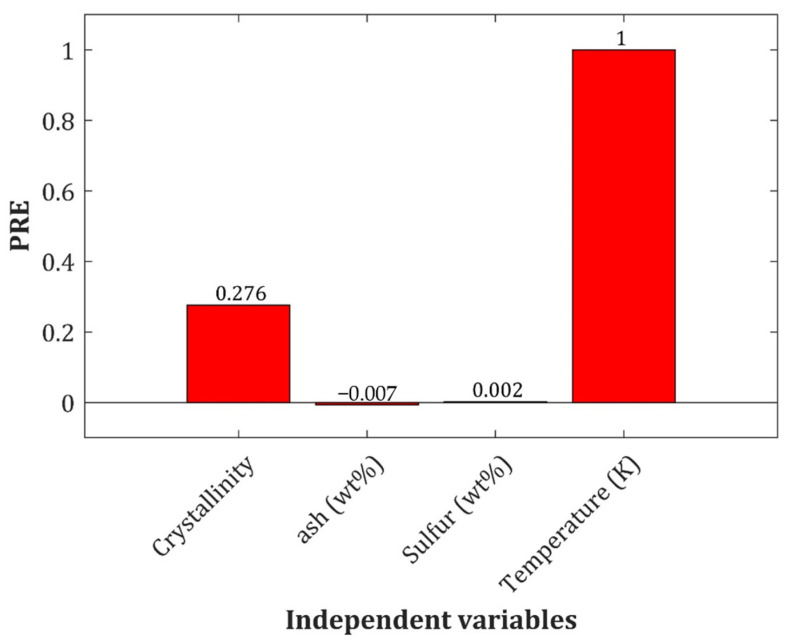
The relative importance of each feature on the prediction accuracy of the biomass heat capacity.

**Figure 3 molecules-27-06540-f003:**
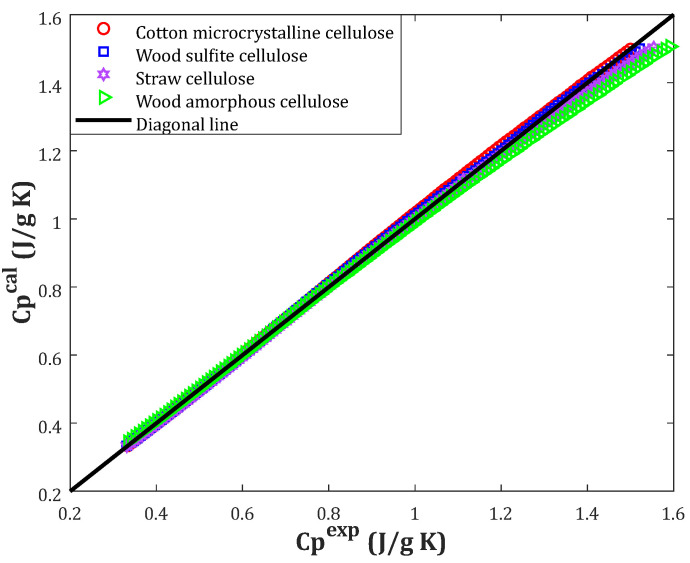
Correlation between actual and estimated heat capacities.

**Figure 4 molecules-27-06540-f004:**
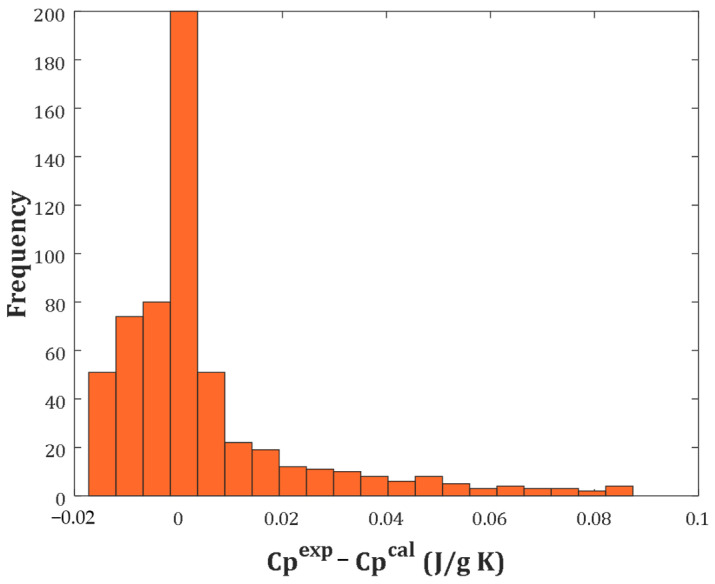
Histogram of the residual error between actual and predicted heat capacities.

**Figure 5 molecules-27-06540-f005:**
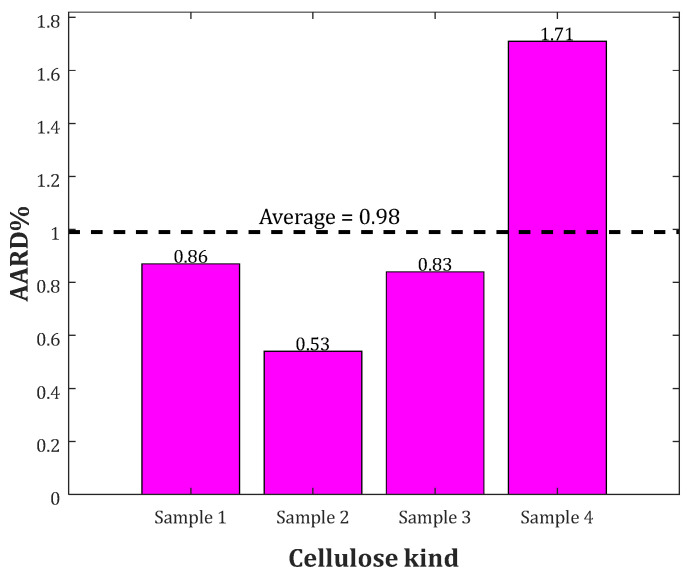
The AARD% associated with the prediction of the heat capacity of each biomass kind.

**Figure 6 molecules-27-06540-f006:**
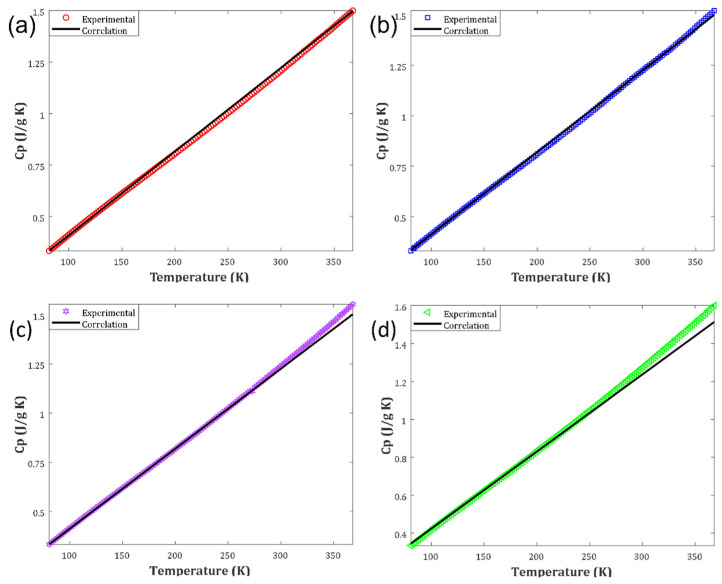
(**a**) Performance of the developed correlation for calculating the Cp versus temperature profile of the cotton microcrystalline cellulose. (**b**) The calculated and experimental profiles of the Cp versus temperature of the wood sulfite cellulose. (**c**)Investigating the temperature effect on the straw cellulose heat capacity from the experimental and modeling observations. (**d**) Performance of the developed correlation for approximating the Cp dependency of the wood amorphous cellulose on the temperature.

**Figure 7 molecules-27-06540-f007:**
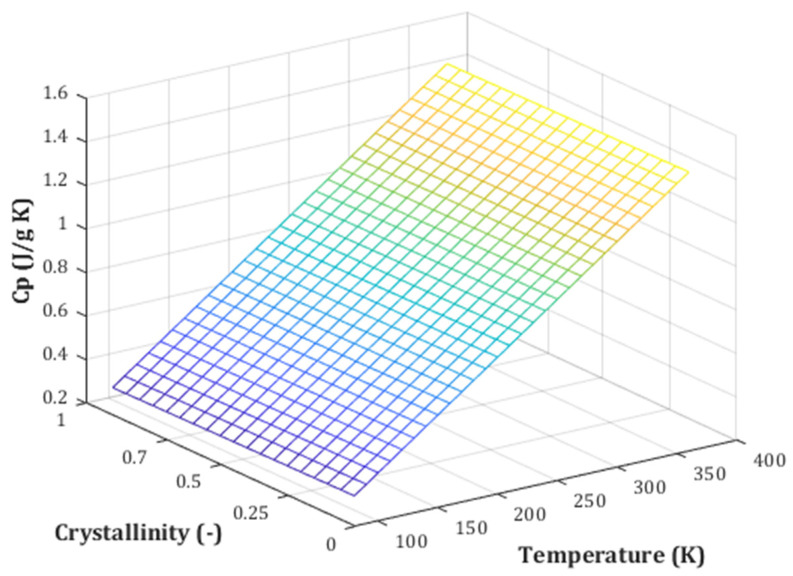
Color map plot of heat capacity of biomass based on temperature and crystallinity index variations.

**Figure 8 molecules-27-06540-f008:**
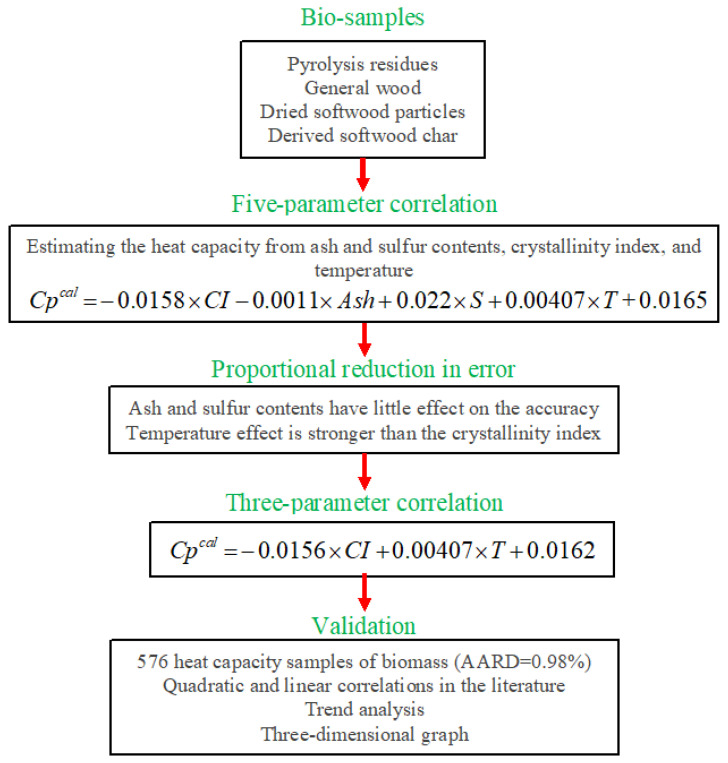
A flowchart of suggested methodology for estimating the biomass heat capacity.

**Table 1 molecules-27-06540-t001:** Adjusted coefficients and AARD% of the linear and quadratic correlations developed based on the temperature.

Correlation	A	B	C	AARD%
Linear	0	0.00406	0.0061	1.29
Quadratic	6.11 × 10^−5^	−0.0210	2.232	39.19

**Table 2 molecules-27-06540-t002:** The heat capacity versus biomass chemical composition, crystallinity index and temperature [32].

Biomass Type	CI (-)	Temperature (K)	Ash (wt%)	S (wt%)	Cp (J/g∙K)	Number Data
Sample 1	0.90	81.50–367.50	0.10	0.02	0.3335–1.500	143
Sample 2	0.80	80.73–367.40	0.10	0.43	0.3304–1.521	144
Sample 3	0.74	80.53–368.25	0.49	0.11	0.3314–1.554	145
Sample 4	0	80.61–368.09	0.07	0.02	0.3342–1.602	144

**Table 3 molecules-27-06540-t003:** Developed correlations in the literature to estimate biomass heat capacity.

Material	Correlation Shape	Temperature	Ref.
Various biomass	Cp = 0.00534 × T − 299	40–80 °C	[22]
Pyrolysis residues	Cp = 0.0014 × T + 688	40–80 °C	[22]
General wood	Cp = 0.0046 × T − 0.113	0–100 °C	[35]
Dried softwood particles	Cp = 0.00546 × T − 0.524	40–140 °C	[36]
Derived softwood char	$Cp\;=\;-\;3.8\times 10{\;}^{-\;6}\;\times \;T^{2}\;+\;0.00598\;\times \;T\;-\;795$	40–140 °C	[36]

## Data Availability

The study data analyzed in this article can be obtained by request from the corresponding author.

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
