# Peer review of "Introducing a Linear Empirical Correlation for Predicting the Mass Heat Capacity of Biomaterials"

_molecules, 2022, doi:10.3390/molecules27196540_

Round 1
Reviewer 1 Report
Scientific literature is rich of correlation of Cp with temperature that work quite well. Sincerely even if the work is quite well written in the form, It is not clear the real contribute to the scientific community of this model. In other word we have model to calculate heat capacity of biomass as function of temperature. Now the authors are proposing a model to calculate heat capacity as function of temperature, crystallinity, ash, and Sulphur. It would be useful to have a model to calculate heat capacity as function of crystallinity and other variable strictly linked to the heat capacity.
Moreover, I have the following question for the authors:
Is the sulphur content an important variable to determine biomass heat capacity? Why should sulphur modify the heat capacity of biomass? Which is the amount of S in a biomass?
Why do not you consider nitrogen content of biomass that is higher than sulphur?
As concerns the ash content the authors consider only biomass with low ash content. E.g. why do not you consider straw or digestate? In these cases, the influence of ash on the specific heat would not be neglectable. How does work the model with the straw?
Author Response
Reviewer 1:
Scientific literature is rich of correlation of Cp with temperature that work quite well. Sincerely even if the work is quite well written in the form, It is not clear the real contribute to the scientific community of this model. In other word we have model to calculate heat capacity of biomass as function of temperature. Now the authors are proposing a model to calculate heat capacity as function of temperature, crystallinity, ash, and Sulphur. It would be useful to have a model to calculate heat capacity as function of crystallinity and other variable strictly linked to the heat capacity.
Dear Reviewer 1
We would like to thank you for your constructive comments and the time put into reviewing the manuscript. We believe the manuscript has substantially improved after carefully addressing your valuable comments and making the suggested modifications. The corresponding changes are summarized in our response below and highlighted in yellow in the revised manuscript.
We hope our revisions have improved the paper quality to your satisfaction level.
1) Is the sulphur content an important variable to determine biomass heat capacity? Why should sulphur modify the heat capacity of biomass? Which is the amount of S in a biomass?
Response: As you certainly know, the heat capacity of solid/liquid materials is a function of composition and temperature. Since the source articles of our experimental databank have only measured/reported the ash and sulfur contents of bio-samples, our models investigate the effect of these ingredients on biomass heat capacity.
Meanwhile, the proportional reduction in error (REE) approves that ash and sulfur contents have little effect on biomass heat capacity. Therefore, the final correlation estimates the biomass heat capacity from the crystallinity index and temperature and ignores the ash and sulfur contents.
The sulfur content of biomass samples ranges from 0.02 to 0.41 weight percent (see Table 1).
2) Why do not you consider nitrogen content of biomass that is higher than sulphur?
Response:
We completely agree with you that the nitrogen content of biomass is higher than its sulfur. Unfortunately, the source articles of our experimental databank report no information about the biomass composition other than the ash and sulfur content. Therefore, analyzing the nitrogen content’s effect on biomass heat capacity was impossible.
3) As concerns the ash content the authors consider only biomass with low ash content. E.g., why do not you consider straw or digestate? In these cases, the influence of ash on the specific heat would not be neglectable. How does work the model with the straw?
Response: As you certainly know, any empirical/intelligent approach has a range of applications. Therefore, our constructed empirical correlations are solely applicable for estimating the heat capacity of biomass samples covering the range of experimental databank.
The newly added explanations to the revised manuscript (just after Table 1) have declared the following statement:
It should be mentioned that all constructed correlations in this study are only valid for estimating the heat capacity of those bio-samples with the composition listed in Table 1 in the temperature range of 80.53 to 368.25 K.
Since the source articles of our experimental databank report no information about the straw or digestate, our correlations are not able to analyze the effect of this variable on biomass heat capacity.
The following added texts to the manuscript to suggest your valuable comment as an interesting research topic in this field (please see the conclusion section).
Since this study estimated the heat capacity of biomass samples with low ash content, it is a good idea to consider straw/digestate as the independent variable in future research in this field. In these cases, the influence of ash on the specific heat capacity would not be neglectable.

Reviewer 2 Report
1) Provide Results in Abstract
2) It would be better if authors provide a flow chart for methodology
3) Line 61 mentions "combustion/gasification/pyrolysis" enabling products. Combustion word can be removed in this context
4) Line 69 mentions " Heat transfer inside the bio-molecules" and same can be worded as "molecular kinetics" as the cause of thermal activity.
Author Response
Reviewer 2:
Dear Reviewer 2
Thank you very much for the positive feedback. We wish to thank you for reviewing the paper. All your comments have been addressed within the context of the article. Your comments have definitely led to improving the quality of the revised manuscript. The corresponding changes made in the revised paper are summarized in our response below and highlighted in yellow color in the manuscript body.
We hope our revisions have improved the paper quality to your satisfaction level.
1) Provide Results in Abstract
Response: Some key numerical results have been added to the abstract of the revised manuscript.
2) It would be better if the authors provide a flow chart for methodology
Response: The revised manuscript has addressed this comment in a newly added section (please see section 3.6. Summary of the study in the flowchart form).
3) Line 61 mentions "combustion/gasification/pyrolysis" enabling products. Combustion words can be removed in this context
Response: Thank you so much for this suggestion. The revised manuscript has considered this comment.
4) Line 69 mentions "Heat transfer inside the bio-molecules" and the same can be worded as "molecular kinetics" as the cause of thermal activity.
Response: Thank you so much for this suggestion. The revised manuscript has considered this comment.

Round 2
Reviewer 1 Report
I appreciate your efforts to improve your work according my suggestion. The range of applicability of the was defined.